# The Association between Family History of Lung Cancer and Development of Lung Cancer: Analysis from the KoGES Data in Korea

**DOI:** 10.3390/cancers16112063

**Published:** 2024-05-30

**Authors:** Sang Hyuk Kim, Hyun Lee, Bo-Guen Kim, Sang-Heon Kim, Jang Won Sohn, Ho Joo Yoon, Seung Hun Jang, Dong Won Park

**Affiliations:** 1Division of Pulmonary, Allergy and Critical Care Medicine, Department of Internal Medicine, Dongguk University Gyeongju Hospital, Dongguk University College of Medicine, Gyeongju 38066, Republic of Korea; gost702@naver.com; 2Division of Pulmonary Medicine and Allergy, Department of Internal Medicine, Hanyang University College of Medicine, Seoul 04763, Republic of Korea; namuhanayeyo@naver.com (H.L.); kbg1q2w3e@gmail.com (B.-G.K.); sangheonkim@hanyang.ac.kr (S.-H.K.); jwsohn@hanyang.ac.kr (J.W.S.); hjyoon@hanyang.ac.kr (H.J.Y.); 3Division of Pulmonary, Allergy and Critical Care Medicine, Department of Medicine, Hallym University Sacred Heart Hospital, Hallym University College of Medicine, Anyang 14068, Republic of Korea

**Keywords:** lung cancer, lung neoplasms, family health history, risk factors, genetic phenomena

## Abstract

**Simple Summary:**

A family history of lung cancer has been reported to increase the risk of lung cancer development. Since a family history of lung cancer is generally regarded as an unmodifiable factor, understanding the relationship between a family history of lung cancer and the development of lung cancer is essential for establishing lung cancer screening strategies. However, previous studies on this issue did not consider various characteristics of study participants, such as age, sex, and smoking status, and there is an ongoing debate regarding which subgroups exhibit a higher genetic predisposition to lung cancer. Furthermore, no data are available on the impact of a family history of lung cancer on lung cancer risk in the general Korean population. Therefore, we aimed to investigate the association between a family history of lung cancer in first-degree relatives and lung cancer development through a comprehensive analysis of a large population-based cohort in Korea.

**Abstract:**

Comprehensive analyses of the association between a family history of lung cancer and lung cancer risk are limited, especially in the Korean population. We used baseline data from the Korean Genome and Epidemiology Study, conducted between 2001 and 2013. This study enrolled 198,980 individuals. Lung cancer diagnoses and family histories were determined using questionnaires. Multivariable logistic regression analysis was performed to evaluate the effect of family history on the risk of lung cancer. Of 198,980 individuals, 6296 (3.2%) and 140 (0.1%) had a family history of lung cancer and lung cancer, respectively. Individuals with a family history of lung cancer in first-degree relatives (FDRs) had a higher risk of lung cancer development than those without (adjusted odds ratio [aOR] = 2.28, 95% confidence interval [CI] = 1.11–4.66). This was more pronounced in young individuals (<60 years) who had affected relatives diagnosed with lung cancer before the age of 60 years (aOR = 3.77, 95% CI = 1.19–11.88). In subgroup analyses, this association was more evident in women, never smokers, and young individuals. A family history of lung cancer, especially in FDRs, is a significant risk factor for lung cancer development in Korea.

## 1. Background

Lung cancer is responsible for the majority of cancer-related deaths worldwide [1]. Although smoking is considered the primary driver of most lung cancers [2], there are other risk factors associated with lung cancer development, including exposure to radiation, environmental contaminants, occupational carcinogens, and a family history of the disease [3,4,5]. Among these, a family history of lung cancer is generally regarded as an unmodifiable factor, highlighting the importance of early detection and regular screening for those at increased risk [6]. Therefore, understanding the relationship between a family history of lung cancer and the development of lung cancer is essential for establishing lung cancer screening strategies.

The effect of family history on lung cancer risk can be affected by several factors, such as ethnicity, degree of relatives, and the age at which family members are diagnosed with lung cancer [7,8]. A more in-depth analysis that incorporates these variables is required to understand the role of family history in lung cancer development. However, most studies that have considered these factors have been primarily case-control studies, which have limited ability to draw conclusions from the general population [7,8,9]. A longitudinal U.S. population-based study found that participants with a familial history of lung cancer had an increased risk of developing lung cancer, with a focus on the degree and age of affected relatives at diagnosis [10]. However, this study did not consider other characteristics, such as age, sex, and smoking status. There is an ongoing debate regarding which subgroups exhibit a higher genetic predisposition to lung cancer. Furthermore, no data are available on the impact of a family history of lung cancer on lung cancer risk in the general Korean population.

Therefore, we aimed to investigate the association between a family history of lung cancer in first-degree relatives (FDRs) and lung cancer development, focusing on the degree and age at diagnosis of affected relatives, through a comprehensive analysis of data from a large population-based cohort in Korea.

## 2. Methods

### 2.1. Data Source and Study Population

The Korean Genome and Epidemiology Study (KoGES) is a community-based longitudinal cohort study conducted by the Korean National Institute of Health, the Korea Centers for Disease Control and Prevention, and the Ministry of Health and Welfare of Korea to assess factors affecting the incidence of chronic diseases. For the initial recruitment, eligible participants were invited to volunteer through a variety of methods, including on-site invitations, mailed letters, telephone calls, media campaigns, or community leader-mediated conferences. Participants were invited to visit the survey sites, which included 50 or more national and international medical schools, hospitals, and health institutions. At these sites, they underwent interviews, completed questionnaires (Appendix A) including family history of cancers administered by trained staff, and received physical examinations. Based on these datasets, several high-quality studies have been conducted [11,12]. More detailed information on KoGES studies were provided in previous studies [13].

Among the KoGES studies, we used the Ansan-Anseong, Health Examinee (HEXA), and Cardiovascular Disease Association (CAVAS) cohorts. These cohorts included community-dwellers aged 40 years or older at the time of enrollment and participants recruited from the national health examinee registry. In our study, we adopted a cross-sectional approach to maximize the number of participants based on baseline data. Baseline measurements were conducted during the following period: 2001–2002 for the Ansan-Anseong study; 2004–2013 for the HEXA study; and 2005–2011 for the CAVAS study.

A total of 211,562 individuals were enrolled for the baseline measurements. Of these, we excluded 10,914 individuals without data on a family history of lung cancer and 1668 individuals with missing values in smoking status. Finally, 198,980 individuals were included in the analytical cohort (Figure 1).

The ethics committee of Hanyang University Hospital (application no. HYUH 2022-08-042) approved the use of the KoGES database. The institutional review board waived the requirement for written informed consent.

### 2.2. Family History of Lung Cancer and Lung Cancer Development

Family history of lung cancer and lung cancer development were assessed using questionnaires. FDRs consisted of parents and siblings, and separate questionnaires were used to inquire about family history, e.g., ‘Do you have a family history of lung cancer in your parents or siblings?’. A family history of lung cancer was further subdivided into three categories based on the relationship between the enrolled individuals and their affected relatives: (1) total, (2) parents, and (3) siblings. To evaluate the impact of age on lung cancer development, we investigated the age at which enrolled individuals and affected relatives were diagnosed with lung cancer using the questionnaire: ‘When were you diagnosed with lung cancer?’ In cases where multiple family members had been diagnosed with lung cancer, we defined ‘having affected relatives aged less than 60 years’ as at least one family member was diagnosed with lung cancer before 60 years.

### 2.3. Covariates

Body mass index (BMI) was calculated by dividing the weight (kg) by the square of the height (m^2^). BMI was categorized as follows: underweight (BMI < 18.5 kg/m^2^), normal weight (BMI 18.5–24.9 kg/m^2^), overweight or obese (BMI ≥25) kg/m^2^. Smoking history was assessed using questionnaires and classified as never smoker and ever smoker. Income was determined based on monthly household income and categorized into the lowest (1st quartile), middle (2nd and 3rd quartile), and highest (4th quartile). Marital status was assessed using a questionnaire and categorized into never married, married, and divorced or separated. Comorbidities (hypertension, diabetes mellitus, and dyslipidemia) were defined by physician diagnoses using questionnaires.

### 2.4. Statistical Analyses

We compared the individuals according to the presence or absence of lung cancer and a family history of lung cancer. Mean ± standard deviation (SD) and number with percentage were used to express continuous and categorical variables, respectively. We used a *t*-test for continuous variables and a χ-squared test for categorical variables. Since this was a cross-sectional study, we calculated the prevalence of lung cancer and odds ratio of lung cancer development. To evaluate the odds of lung cancer development, a multivariable logistic regression analysis was performed with adjustments for age, sex, and smoking status. The interaction between the variables used to stratify the group and family history of lung cancer was determined by adding an interaction term to the multivariable logistic regression analysis. A stratified analysis was conducted based on age, sex, and smoking status. Statistical significance was determined using a two-sided *p*-value < 0.05. R software version 4.2.2 (R core Team 2019; R Foundation for Statistical Computing, Vienna, Austria) was used for all statistical analyses.

## 3. Results

### 3.1. Baseline Characteristics

The baseline characteristics of the enrolled individuals are shown in Table 1. Of the 198,980 individuals, the prevalence of lung cancer and a family history of lung cancer was 0.1% and 3.2%, respectively. Individuals with lung cancer were older (62.4 ± 7.2 years vs. 53.9 ± 8.7 years, *p* < 0.001), with a higher proportion of men (63.6% vs. 34.8%, *p* < 0.001) compared to those without lung cancer. Additionally, individuals with lung cancer were more likely to be ever smokers (57.9% vs. 27.5%, *p* < 0.001) and had lower income (45.7% vs. 25.8%, *p* < 0.001) and diabetes mellitus (14.3% vs. 7.1%, *p* = 0.002). Regarding a family history of lung cancer, the proportions of individuals with a family history of lung cancer (5.7% vs. 3.1%, *p* = 0.134) and a family history of lung cancer in parents (0.7% vs. 2.5%, *p* = 0.287) were similar between two groups. However, the proportion of siblings with a family history of lung cancer was significantly higher in individuals with lung cancer than in those without lung cancer (5.0% vs. 0.7%, *p* < 0.001). No significant differences were observed in BMI (23.6 ± 2.9 kg/m^2^ vs. 24.0 ± 3.0 kg/m^2^, *p* = 0.135), marital status (86.4% vs. 87.3% for married, *p* = 0.392), and hypertension (26.4% vs. 20.4%, *p* = 0.093).

When stratified by the presence or absence of a family history of lung cancer, individuals with a family history of lung cancer were more likely to be under 60 years old (78.7% vs. 71.8%, *p* < 0.001), female (67.4% vs. 65.1%, *p* < 0.001), belong to the high-income group (27.6% vs. 22.7%, *p* < 0.001), and married (89.6% vs. 87.2%, *p* < 0.001). Additionally, they had a lower prevalence of hypertension (17.7% vs. 20.4%, *p* < 0.001), diabetes mellitus (6.3% vs. 7.1%, *p* = 0.025), but had a higher prevalence of dyslipidemia (10.3% vs. 8.7%, *p* < 0.001) (Table 2). However, no significant difference was found in the BMI and smoking status between the two groups (*p* > 0.05 for both).

### 3.2. Family History of Lung Cancer and Lung Cancer Development

As shown in Table 3, individuals with a family history of lung cancer in FDRs had higher odds for lung cancer development than those without (adjusted odds ratio (aOR) = 2.28, 95% confidence interval (CI) = 1.11–4.66). A family history of lung cancer significantly increased the risk of lung cancer development in individuals who had affected relatives diagnosed with lung cancer before the age of 60 years (aOR = 3.77, 95% CI = 1.19–11.88). However, this association was not statistically significant in those who had affected relatives diagnosed with lung cancer after the age of 60 years (aOR = 1.84, 95% CI = 0.75–4.50).

When stratified by age, the impact of a family history of lung cancer was more pronounced in individuals aged < 60 years who had affected relatives diagnosed with lung cancer before the age of 60 years (aOR = 5.89, 95% CI = 1.42–24.39). Among women and never smokers, the increased risk of lung cancer development was more evident in individuals with a family history of lung cancer who had affected relatives diagnosed with lung cancer before the age of 60 years (aOR = 9.26, 95% CI = 2.87–29.91 for women; aOR = 8.52, 95% CI = 2.65–27.39 for never smokers). However, a family history of lung cancer was not significantly associated with lung cancer development in men and ever smokers (aOR = 1.41, 95% CI = 0.44–4.48 for men; aOR = 1.50, 95% CI = 0.47–4.78 for ever smokers). There was no interaction effect for age, sex, or smoking status (*p* for interaction > 0.05 for all).

## 4. Discussion

In this study, we evaluated the impact of a family history of lung cancer in terms of the degree of relatives and age at diagnosis of affected relatives on lung cancer development using a Korean population-based large-scale cohort. The important findings of this study are as follows. First, within the general Korean population, the prevalence of a family history of lung cancer in FDRs was 3%. Second, a notable correlation was found between a family history of lung cancer in FDRs and an increased risk of lung cancer development in the Korean population. Third, the impact of a family history of lung cancer was more evident in young individuals (<60 years) who had affected relatives diagnosed with lung cancer before the age of 60 years. Fourth, young individuals, women, and never smokers had higher odds of lung cancer development when they had a family history of lung cancer.

Recent meta-analyses have identified a significant association between a family history of lung cancer and an increased risk of lung cancer development [14]. However, most of the studies were case-control designs [7,15,16,17,18], and none of the remaining population-based studies were conducted in the Asian population [10,19,20,21]. We found a 2.3-fold increase in the risk of developing lung cancer in individuals with a family history of lung cancer in the Asian population using a population-based cohort. Our data strongly underline the possibility of a familial history of lung cancer as a significant risk factor for lung cancer in a broad Asian population.

Our study also highlighted the importance of age at the diagnosis of lung cancer in the affected relatives in terms of the impact of a family history of lung cancer on the development of lung cancer. Notably, this association was particularly pronounced in young individuals. These results support previous findings that earlier lung cancer diagnosis of affected relatives indicates a genetic predisposition to lung cancer [7,8,22] and further suggest that young adults were more prone to this type of lung cancer. Another important finding was that women and never smokers with a family history of lung cancer showed higher odds of lung cancer development than men and those who had smoked. Several studies have consistently shown a stronger connection between a family history of lung cancer and lung cancer development in women and never smokers [23,24,25]. Considering our findings, in conjunction with results of these studies, these groups may be more susceptible to lung cancer development owing to their family history.

Familial cancers, including lung cancer, are known to be diagnosed earlier than sporadic cancers [26,27]. Despite the dominant role of environmental factors, our study highlights the theory of genetic predisposition to lung cancer [28]. In addition, it is well established that patterns of lung cancer development vary by sex, race, and smoking status [29,30]. Our study also found disparities in the impact of a family history of lung cancer on lung cancer development across sexes and smoking status. Therefore, the current lung cancer screening guidelines for those aged > 54 years who satisfy the smoking history criterion may not be sufficient for all those at a higher lung cancer risk [31]. More proactive strategies are needed to identify high-risk groups outside of the current guidelines. These risk-reduction strategies could include regular lung cancer screening and surveillance using computed tomography, smoking cessation programs, genetic counseling, and lifestyle modifications.

Our study had some limitations. First, the baseline data of KoGES studies were cross-sectional, limiting the ability to establish causal inferences between family history and lung cancer development. Second, the number of lung cancer cases was relatively small, which could lead to statistical insignificance and limit the descriptive analyses. Third, this study has lacked environmental information. Therefore, individuals with lung cancer and their affected family members may have lived close to each other, potentially leading to similar exposure to environmental carcinogens. Fourth, the prevalence of lung cancer may be underestimated, as some patients may have died, and severely ill lung cancer patients might not have participated in the KoGES study. Fifth, recall bias may be present, given the use of survey data. Sixth, the generalizability of our findings should be cautiously examined as all enrolled individuals were from a single Asian country.

## 5. Conclusions

In conclusion, a family history of lung cancer, especially FDRs, is a significant risk factor for lung cancer development in Korea. Identifying a family history of lung cancer may contribute to early detection in populations susceptible to genetic lung cancer, including young adults, females, never smokers, and people who have affected relatives diagnosed with lung cancer before the age of 60 years. Intensive screening and risk-reducing strategies should be considered for populations with a genetic predisposition to lung cancer.

## Figures and Tables

**Figure 1 cancers-16-02063-f001:**
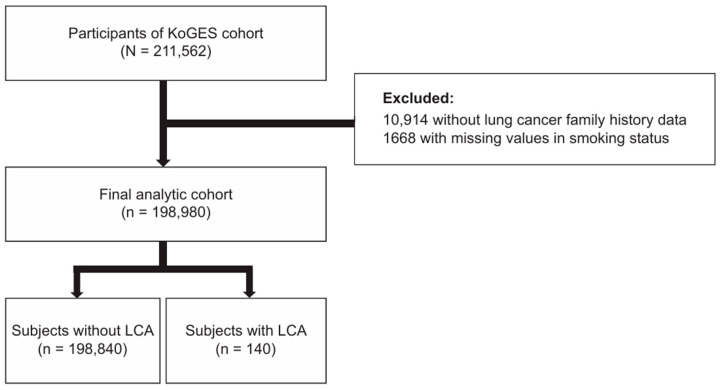
Flow chart of the study population. Abbreviations: KoGES, Korean Genome and Epidemiology; LCA, lung cancer.

**Table 1 cancers-16-02063-t001:** Clinical characteristics of the study population.

	Total (N = 198,980)	Individuals without Lung Cancer (n = 198,840)	Individuals with Lung Cancer (n = 140)	*p*-Value
**Age, years**	53.9 ± 8.7	53.9 ± 8.7	62.4 ± 7.2	<0.001
40–49 years	68,359 (34.4)	68,353 (34.4)	6 (4.3)	
50–59 years	74,860 (37.6)	74,822 (37.6)	38 (27.1)	<0.001
60–69 years	48,853 (24.6)	48,774 (24.5)	79 (56.4)	
70–79 years	6651 (3.3)	6635 (3.3)	16 (11.4)	
≥80 years	257 (0.1)	256 (0.1)	1 (0.7)	
**Gender**				<0.001
Female	129,649 (65.2)	129,598 (65.2)	51 (36.4)	
Male	69,331 (34.8)	69,242 (34.8)	89 (63.6)	
**Body mass index, kg/m^2^ (n = 198,119)**	24.0 ± 3.0	24.0 ± 3.0	23.6 ± 2.9	0.135
<18.5 kg/m^2^	3590 (1.8)	3586 (1.8)	4 (2.9)	0.627
18.5–24.9 kg/m^2^	126,973 (64.1)	126,882 (64.1)	91 (65.5)	
≥25 kg/m^2^	67,556 (34.1)	67,512 (34.1)	44 (31.7)	
**Smoking status**				<0.001
Never smoker	144,291 (72.5)	144,232 (72.5)	59 (42.1)	
Ever smoker	54,689 (27.5)	54,608 (27.5)	81 (57.9)	
**Income (n = 159,381)**				<0.001
Lowest	41,098 (25.8)	41,045 (25.8)	53 (45.7)	
Middle	81,894 (51.4)	81,842 (51.4)	52 (44.8)	
Highest	36,389 (22.8)	36,378 (22.8)	11 (9.5)	
**Marital status (n = 198,011)**				0.392
Never married	4052 (2.0)	4051 (2.0)	1 (0.7)	
Married	172,881 (87.3)	172,760 (87.3)	121 (86.4)	
Divorced or separated	21,078 (10.6)	21,060 (10.6)	18 (12.9)	
**Comorbidities**				
Hypertension (n = 198,863)	40,485 (20.4)	40,448 (20.4)	37 (26.4)	0.093
Diabetes mellitus (n = 198,812)	14,037 (7.1)	14,017 (7.1)	20 (14.3)	0.002
Dyslipidemia (n = 131,188)	17,336 (8.7)	17,315 (8.7)	21 (15.0)	0.013
**Lung cancer family history**				
Total	6296 (3.2)	6288 (3.2)	8 (5.7)	0.138
Parents	4938 (2.5)	4937 (2.5)	1 (0.7)	0.283
Siblings	1436 (0.7)	1429 (0.7)	7 (5.0)	<0.001

Data are expressed as a mean ± standard deviation for continuous variables and a number (percentage) for categorical variables.

**Table 2 cancers-16-02063-t002:** Clinical characteristics of the study population according to the presence or absence of the family history of lung cancer.

	Individuals without a Family History of Lung Cancer (n = 192,684)	Individuals with a Family History of Lung Cancer (n = 6296)	*p*-Value
**Age, years**			<0.001
<60 years	138,262 (71.8)	4957 (78.7)	
≥60 years	54,422 (28.2)	1339 (21.3)	
**Gender**			<0.001
Female	125,405 (65.1)	4244 (67.4)	
Male	67,279 (34.9)	2052 (32.6)	
**Body mass index, kg/m^2^ (n = 198,119)**			
<18.5 kg/m^2^	3484 (1.8)	106 (1.7)	0.067
18.5–24.9 kg/m^2^	122,866 (64.0)	4107 (65.5)	
≥25 kg/m^2^	65,495 (34.1)	2061 (32.8)	
**Smoking status**			0.546
Never smoker	139,747 (72.5)	4544 (72.2)	
Ever smoker	52,937 (27.5)	1752 (27.8)	
**Income (n = 159,381)**			
Lowest	40,012 (26.0)	1086 (20.3)	
Middle	79,103 (51.4)	2791 (52.1)	<0.001
Highest	34,911 (22.7)	1478 (27.6)	
**Marital status (n = 198,011)**			
Never married	3919 (2.0)	133 (2.1)	
Married	167,251 (87.2)	5630 (89.6)	<0.001
Divorced or separated	20,555 (10.7)	523 (8.3)	
**Comorbidities**			
Hypertension (n = 198,863)	39,369 (20.4)	1116 (17.7)	<0.001
Diabetes mellitus (n = 198,812)	13,638 (7.1)	399 (6.3)	0.025
Dyslipidemia (n = 131,188)	16,689 (8.7)	647 (10.3)	<0.001

Data are expressed as a number (percentage).

**Table 3 cancers-16-02063-t003:** Association between family history of lung cancer and lung cancer development.

Family History of Lung Cancer	Lung Cancer, n (%)	Outcome: Lung Cancer	
		Unadjusted OR (95% CI, *p*)	Adjusted OR (95% CI, *p*)
**Total population**			
No (n = 192,684)	132 (0.07)	Ref.	Ref.
Yes (n = 6296)	8 (0.13)	1.86 (0.91–3.79, 0.090)	2.28 (1.11–4.66, 0.024)
Age of affected relatives < 60 years (n = 1436)	3 (0.21)	3.05 (0.97–9.60,0.056)	3.77 (1.19–11.88,0.024)
Age of affected relatives ≥ 60 years (n = 4860)	5 (0.10)	1.50 (0.61–3.67, 0.372)	1.84 (0.75–4.50, 0.182)
**Age < 60 years**			
No (n = 138,262)	41 (0.03)	Ref.	Ref.
Yes (n = 4957)	3 (0.06)	2.04 (0.63–6.59, 0.233)	2.03 (0.63–6.56, 0.237)
Age of affected relatives < 60 years (n = 1147)	2 (0.17)	5.89 (1.42–24.37, 0.014)	5.89 (1.42–24.39, 0.015)
Age of affected relatives ≥ 60 years (n = 3810)	1 (0.03)	0.89 (0.12–6.44, 0.904)	0.88 (0.12–6.39, 0.898)
**Age ≥ 60 years**			
No (n = 54,422)	91 (0.17)	Ref.	Ref.
Yes (n = 1339)	5 (0.37)	2.24 (0.91–5.51, 0.080)	2.42 (0.98–5.98, 0.055)
Age of affected relatives < 60 years (n = 289)	1 (0.35)	2.07 (0.29–14.93, 0.469)	2.13 (0.30–15.40, 0.453)
Age of affected relatives ≥ 60 years (n = 1050)	4 (0.38)	2.28 (0.84–6.23, 0.107)	2.50 (0.92–6.85, 0.074)
*p* for interaction *			0.790
**Men**			
No (n = 67,279)	86 (0.13)	Ref.	Ref.
Yes (n = 2052)	3 (0.15)	1.14 (0.36–3.62, 0.819)	1.41 (0.44–4.48, 0.560)
Age of affected relatives < 60 years (n = 433)	0	-	-
Age of affected relatives ≥ 60 years (n = 1619)	3 (0.19)	1.45 (0.46–4.59, 0.527)	1.83 (0.57–5.81, 0.307)
**Women**			
No (n = 125,405)	46 (0.04)	Ref.	Ref.
Yes (n = 4244)	5 (0.12)	3.21 (1.28–8.09, 0.013)	3.58 (1.42–9.02, 0.007)
Age of affected relatives < 60 years (n = 1003)	3 (0.30)	8.18 (2.54–26.33, <0.001)	9.26 (2.87–29.91, <0.001)
Age of affected relatives ≥ 60 years (n = 3241)	2 (0.06)	1.68 (0.41–6.93, 0.471)	1.86 (0.45–7.69, 0.390)
*p* for interaction *			0.174
**Ever smoker**			
No (n = 52,937)	78 (0.15)	Ref.	Ref.
Yes (n = 1752)	3 (0.17)	1.16 (0.37–3.69, 0.798)	1.50 (0.47–4.78, 0.492)
Age of affected relatives < 60 years (n = 371)	0	-	-
Age of affected relatives ≥ 60 years (n = 1381)	3 (0.22)	1.48 (0.47–4.68, 0.509)	1.95 (0.61–6.21, 0.259)
**Never smoker**			
No (n = 139,747)	54 (0.04)	Ref.	Ref.
Yes (n = 4544)	5 (0.11)	2.85 (1.14–7.13, 0.025)	3.25 (1.30–8.16, 0.012)
Age of affected relatives < 60 years (n = 1065)	3 (0.28)	7.31 (2.28–23.41, < 0.001)	8.52 (2.65–27.39, < 0.001)
Age of affected relatives ≥ 60 years (n = 3479)	2 (0.06)	1.49 (0.36–6.11, 0.581)	1.69 (0.41–6.94, 0.467)
*p* for interaction *			0.251

* The value of *p* for interaction was evaluated for the interaction between subgroup category and family history of lung cancer (no vs. yes) for lung cancer development without consideration of the age of the affected relative. Age, sex, and smoking status were included for the adjustment. Abbreviations: OR, odds ratio; CI, confidence interval.

## Data Availability

The datasets analyzed during the study are not publicly available due to personal data protection. However, they can be obtained from the corresponding author upon reasonable request.

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
