# Peer review of "The Association between Family History of Lung Cancer and Development of Lung Cancer: Analysis from the KoGES Data in Korea"

_cancers, 2024, doi:10.3390/cancers16112063_

Round 1
Reviewer 1 Report
Comments and Suggestions for Authors
Kim et al. investigated the association between a family history of lung cancer and lung cancer risk using data from the Korean Genome and Epidemiology Study. Individuals with a family history of lung cancer in first-degree relatives (FDRs) had a higher risk of lung cancer development than those without, especially young individuals, women, and never smokers.
The article is interesting. The main limitation is the fact that the relationship between a family cancer history and cancer risk is already well investigated in the past, even when not necessary in Korea. Further limitation is that people with lung cancer often die within some months. Authors compared the individuals according to the presence or absence of lung cancer. The prevalence of lung cancer may be strongly underestimated as people with lung lancer who were severe ill or died, had lower chance to take part on the questionnaire in the past. People with lung cancer in this study were 63 years old. They are usually over seventy in average. Authors should discuss that in more details.
Regarding methods, authors write “Multivariable logistic analysis”. Correctly, this is multivariable logistic regression analysis”.
Author Response
## Response to Reviewer 1’s comments
General Comments (G). Kim et al. investigated the association between a family history of lung cancer and lung cancer risk using data from the Korean Genome and Epidemiology Study. Individuals with a family history of lung cancer in first-degree relatives (FDRs) had a higher risk of lung cancer development than those without, especially young individuals, women, and never smokers.
Response (R). Thank you for your thorough summary and positive feedback on our study.
Comments 1 (C1). The article is interesting. The main limitation is the fact that the relationship between a family cancer history and cancer risk is already well investigated in the past, even when not necessary in Korea.
Response (R1). Thank you for your valuable comment. We agree that the relationship between family history of cancer and cancer risk has been extensively studied in various populations. However, we believe our study provides another insight specific to the age at diagnosis of lung cancer in both patients and their affected relatives. In the revised manuscript, we have focused more on this point in our discussion.
Page 7, 2nd to 3rd paragraph and page 8, 1st paragraph
"Recent meta-analysis have identified a significant association between a family history of lung cancer and an increased risk of lung cancer development [14]. However, most of the studies were case-control designs [7,15-18], and none of the remaining population-based studies were conducted in the Asian population [10,19-21]. We found a 2.4-fold increase in the risk of developing lung cancer in individuals with a family history of lung cancer in the Asian population using a population-based cohort. Our data strongly underline the possibility of a familial history of lung cancer as a significant risk factor for lung cancer in a broad Asian population.
Our study also highlighted the importance of age at the diagnosis of lung cancer in the affected relatives in terms of the impact of a family history of lung cancer on the development of lung cancer. Notably, this association was particularly pronounced in young individuals. These results support previous findings that earlier lung cancer diagnosis of affected relatives indicates a genetic predisposition to lung cancer [7,8,22], and further suggest that young adults were more prone to this type of lung cancer. Another important finding was that women and never smokers with a family history of lung cancer showed higher odds of lung cancer development than men and those who had smoked. Several studies have consistently shown a stronger connection between a family history of lung cancer and lung cancer development in women and never smokers [23-25]. Considering our findings, in conjunction with those of these studies, these groups may be more susceptible to lung cancer development owing to their family history."
(C2). Further limitation is that people with lung cancer often die within some months. Authors compared the individuals according to the presence or absence of lung cancer. The prevalence of lung cancer may be strongly underestimated as people with lung lancer who were severe ill or died, had lower chance to take part on the questionnaire in the past. People with lung cancer in this study were 63 years old. They are usually over seventy in average. Authors should discuss that in more details.
(R2). Thank you for your insightful comment. We agree that this is an important limitation that should be addressed in our discussion. Hence, we further discussed underestimation issues.
Page 13, 1st paragraph
"... Fourth, the prevalence of lung cancer may be underestimated, as some patients may have died, and severely ill lung cancer patients might not have participated in the KoGES study."
(C3). Regarding methods, authors write “Multivariable logistic analysis”. Correctly, this is multivariable logistic regression analysis”.
(R3). Thank you for pointing out the need for more precise terminology in describing our statistical methods. We agree with your suggestion and have revised the text to accurately reflect the correct term, "multivariable logistic regression analysis."
Page 4, abstract
“Multivariable logistic regression analysis was performed to evaluate the effect of family history on the risk of lung cancer.”

Reviewer 2 Report
Comments and Suggestions for Authors
In this article, Kim et al analyzed data from the Korean Genome and Epidemiology Study, conducted between 2001 and 2013. This study enrolled 130,621 individuals. Lung cancer diagnoses and family histories were determined using questionnaires. Multivariable logistic analysis was performed to evaluate the effect of family history on the risk of lung cancer. Of 130,621 individuals, 3,948 (3%) and 134 (0.1%) had a family history of lung cancer and lung cancer, respectively. Individuals with a family history of lung cancer in first-degree relatives (FDRs) had a higher risk of lung cancer development than those without. This was more pronounced in young individuals (< 60 years) who had affected relatives diagnosed with lung cancer before the age of 60 years. In subgroup analyses, this association was more evident in women, never-smokers, and young individuals. They concluded that a family history of lung cancer, especially in FDRs, is a significant risk factor for lung cancer development in Korea.
The authors and others have analyzed a large set of data and the results obtained are considered to be valid. The description of the methods used, the research methodology, and the method of describing the results are well organized and easy to understand.Not only univariate analysis but also multivariate analysis was conducted, and the results are considered reasonable.
Author Response
## Response to Reviewer 2’s comments
General Comments. In this article, Kim et al analyzed data from the Korean Genome and Epidemiology Study, conducted between 2001 and 2013. This study enrolled 130,621 individuals. Lung cancer diagnoses and family histories were determined using questionnaires. Multivariable logistic analysis was performed to evaluate the effect of family history on the risk of lung cancer. Of 130,621 individuals, 3,948 (3%) and 134 (0.1%) had a family history of lung cancer and lung cancer, respectively. Individuals with a family history of lung cancer in first-degree relatives (FDRs) had a higher risk of lung cancer development than those without. This was more pronounced in young individuals (< 60 years) who had affected relatives diagnosed with lung cancer before the age of 60 years. In subgroup analyses, this association was more evident in women, never-smokers, and young individuals. They concluded that a family history of lung cancer, especially in FDRs, is a significant risk factor for lung cancer development in Korea. The authors and others have analyzed a large set of data and the results obtained are considered to be valid. The description of the methods used, the research methodology, and the method of describing the results are well organized and easy to understand. Not only univariate analysis but also multivariate analysis was conducted, and the results are considered reasonable.
(R1). We thank you for your insightful comments and positive evaluation.

Reviewer 3 Report
Comments and Suggestions for Authors
This is the manuscript of a study that examined the association between family history of Lung cancer and development of lung cancer. The results provide added evidence that family history is significantly associated with the development of lung cancer.
There are comments below for author consideration.
Title: This states that this is a prospective cohort study. However, in the limitation, it states that KoGES studies were cross-sectional. These are confusing and require clarification.
Abstract and simple summary: These appear okay.
Background: Revise the sentence "Among these, a family history of lung cancer is generally regarded as an unmodifiable factor that can only be treated with early detection." How do treat family history?
Methods: The description of the data source is scanty and needs to be expanded. Tell the reader more about the data source...Who owns and manages the data? What type of variables are present in the data? How was the questionnaire delivered (in-person, mail, telephone)? What are the response rate and reliability indices of the survey instrument?
There is no mention about ethical considerations. Was IRB approval and informed consent obtained for the study?
What was the reason for excluding individuals that were less than 50 years?
A description of how the variables were categorized is required. For BMI, underweight, normal weight, overweight and obese and their measures. Same for smoking status, income, marital status and comorbidities...Include the categories of these variables and their measures. For family history, what is the meaning of "total". Provide a description and measures of the categories for family history as well.
Results: Calculate and include the percentages of male, female, <60 and >60, years, marital status, and comorbidities, with and without family history of lung cancer.
The tables and flow chart look good.
Discussion: This is mostly okay except for the limitation. Clarify if this is a cross-sectional or prospective cohort study.
Conclusion: Provide some examples of the risk-reduction strategies to be considered for populations with a genetic predisposition to lung cancer.
See highlighted text in pink in the attached file for your convenience in revising the manuscript.
Best of luck!
Comments on the Quality of English LanguageThe quality of English language is okay. Check for minor typos.
Author Response
## Response to Reviewer 3’s comments
General Comments. This is the manuscript of a study that examined the association between family history of Lung cancer and development of lung cancer. The results provide added evidence that family history is significantly associated with the development of lung cancer. There are comments below for author consideration.
(C1). Title: This states that this is a prospective cohort study. However, in the limitation, it states that KoGES studies were cross-sectional. These are confusing and require clarification.
(R1). Thank you for highlighting this important issue. We apologize for the confusion caused by the inconsistent descriptions of the study design. We have changed the title to clarify this matter.
Page 1, title
"The Association Between Family History of Lung Cancer and Development of Lung Cancer: analysis from the KoGES data in Korea"
(C2). Abstract and simple summary: These appear okay.
(R2). We thank you for your insightful comments and positive evaluation.
(C3). Background: Revise the sentence "Among these, a family history of lung cancer is generally regarded as an unmodifiable factor that can only be treated with early detection." How do treat family history?
(R3). Thank you for pointing out the ambiguity in the sentence. We agree that the phrasing needs to be clarified. Family history itself is not a condition that can be treated; rather, it is a risk factor that necessitates proactive management strategies. We revised our manuscript to present the intended meaning more accurately.
Page 5, 1st paragraph
“Among these, a family history of lung cancer is generally regarded as an unmodifiable factor, highlighting the importance of early detection and regular screening for those at increased risk.”
(C4). Methods: The description of the data source is scanty and needs to be expanded. Tell the reader more about the data source...Who owns and manages the data? What type of variables are present in the data? How was the questionnaire delivered (in-person, mail, telephone)? What are the response rate and reliability indices of the survey instrument?
(R4-1). Thank you for your valuable feedback. We appreciate your suggestion to provide a more detailed description of our data source. Our revised manuscript addressed the raised issue in the methods section.
Page 6, 2nd paragraph
"The Korean Genome and Epidemiology Study (KoGES) is a community-based longitudinal cohort study conducted by the Korean National Institute of Health, the Korea Centers for Disease Control and Prevention, and the Ministry of Health and Welfare of Korea to assess factors affecting the incidence of chronic diseases. For the initial recruitment, eligible participants were invited to volunteer through a variety of methods, including on-site invitations, mailed letters, telephone calls, media campaigns, or community leader-mediated conferences. Participants were invited to visit the survey sites, which included 50 or more national and international medical schools, hospitals, and health institutions. At these sites, they underwent interviews, completed questionnaires (Supplementary file 1) including family history of cancers administered by trained staff, and received physical examinations. Based on these datasets, several high-quality studies have been conducted [11,12]. More detailed information on KoGES studies were provided in previous studies [13]."
(C3-2). There is no mention about ethical considerations. Was IRB approval and informed consent obtained for the study?
(R3-2). Thank you for pointing out the need to address ethical considerations in our manuscript. We apologize for this missing and have included a detailed statement on the ethical approval and informed consent process.
Page 7, 3rd paragraph
"The ethics committee of Hanyang University Hospital (application no. HYUH 2022-08-042) approved the use of the KoGES database. The institutional review board waived the requirement for written informed consent."
(C3-3). What was the reason for excluding individuals that were less than 50 years?
(R3-3). Thank you for your insightful comment. The decision to exclude individuals under 50 was based on several considerations, such as lung cancer incidence and the relevance of risk factors. We consider the raised issue by the reviewer and have expanded the analysis, including those under 50.
Page 7, 2nd paragraph
"A total of 211,562 individuals were enrolled for the baseline measurements. Of these, we excluded 10,914 individuals without data on a family history of lung cancerand 1,668 individuals with missing values in smoking status. Finally, 198,980 individuals were included in the analytical cohort (Figure 1)."
(C3-4). A description of how the variables were categorized is required. For BMI, underweight, normal weight, overweight and obese and their measures. Same for smoking status, income, marital status and comorbidities...Include the categories of these variables and their measures. For family history, what is the meaning of "total". Provide a description and measures of the categories for family history as well.
(R3-4). Thank you for your detailed feedback. We appreciate your suggestion to describe how the variables including family history were categorized clearly. We have revised the methods section to include these details.
Page 7, 4th paragraph
"Family history of lung cancer and lung cancer development were assessed using questionnaires. FDRs consisted of parents and siblings, and separate questionnaires were used to inquire about family history, e.g., 'Do you have a family history of lung cancer in your parents or siblings?'. A family history of lung cancer was further subdivided into three categories based on the relationship between the enrolled individuals and their affected relatives: (1) total, (2) parents, and (3) siblings. To evaluate the impact of age on lung cancer development, we investigated the age at which enrolled individuals and affected relatives were diagnosed with lung cancer using the questionnaire: ‘When were you diagnosed with lung cancer?’ In cases where multiple family members had been diagnosed with lung cancer, we defined 'having affected relatives aged less than 60 years' as at least one family member was diagnosed with lung cancer before 60 years."
Page 8, 2nd paragraph
"Body mass index (BMI) was calculated by dividing the weight (kg) by the square of the height (m2). BMI was categorized as follows: underweight (BMI < 18.5 kg/m²), normal weight (BMI 18.5–24.9 kg/m²), overweight or obse (BMI 25.0–29.9 ) kg/m². Smoking history was assessed using questionnaires and classified as never smoker and ever smoker. Income was determined based on monthly household income and categorized into the lowest (1st quartile), middle (2nd and 3rd quartile), and highest (4th quartile). Marital status was assessed by using a questionnaire and categorized into never married, married, and divorced or separated."
(C4-1). Results: Calculate and include the percentages of male, female, <60 and >60, years, marital status, and comorbidities, with and without family history of lung cancer.
(R4-1). Thank you for your insightful comment. We appreciate the suggestion to present more detailed information on the demographics and characteristics. We have revised the results section to include the requested percentages.
Page 10, 1st paragraph
"When stratified by the presence or absence of the family history of lung cancer, individuals with a family history of lung cancer were more likely to be under 60 years old (78.7% vs. 71.8%), female (67.4% vs. 65.1%), belong to the high-income group (27.6% vs. 22.7%), and married (89.6% vs. 87.2%). Additionally, they had a higher prevalence of hypertension (17.7% vs. 20.0%), diabetes mellitus (6.3% vs. 7.1%), and dyslipidemia (10.3% vs. 8.7%) (Table 2). However, no significant difference was found in the BMI and smoking between the two groups (p > 0.05 for both)."
(C4-2). The tables and flow chart look good.
(R4-2). Thank you for your positive feedback regarding the tables and flow chart.
(C5). Discussion: This is mostly okay except for the limitation. Clarify if this is a cross-sectional or prospective cohort study.
(R5). Thank you for your feedback. We apologize for any confusion regarding the study design. To clarify, our study is a cross-sectional analysis of baseline data in the prospective cohort study. We will revise the discussion section to clearly specify this and address the limitations associated with this study design.
Page 13, 1st paragraph
"Our study had some limitations. First, baseline data of KoGES studies were cross-sectional, limiting the ability to establish causal inferences between family history and lung cancer development."
(C6). Conclusion: Provide some examples of the risk-reduction strategies to be considered for populations with a genetic predisposition to lung cancer.
(R6). Thank you for your constructive feedback. We agree that including specific risk-reduction strategies for populations with a genetic predisposition to lung cancer will enhance the practical value of our conclusions. In the revised manuscript, we added examples of risk-reduction strategies.
Page 12, 3rd paragraph and page 13, 1st paragraph
“These risk-reduction strategies could include regular lung cancer screening and surveillance using computed tomography, smoking cessation programs, genetic counseling, and lifestyle modifications.”
(C7). See highlighted text in pink in the attached file for your convenience in revising the manuscript. Best of luck!
(R7). Thank you for your detailed review and for highlighting specific sections in the attached file for revision. We appreciate your guidance in improving the clarity and quality of our manuscript.

Round 2
Reviewer 3 Report
Comments and Suggestions for Authors
The authors have responded adequately to my comments from the initial review and I believe that this revised paper is a much improved version of the initial manuscript.
It was my pleasure participating in the review process.
Best of luck!
Comments on the Quality of English LanguageThe quality of English language is okay.
Check for minor error in typo and grammar.